# Advancing Stroke Research on Cerebral Thrombi with Omic Technologies

**DOI:** 10.3390/ijms24043419

**Published:** 2023-02-08

**Authors:** Gianluca Costamagna, Sara Bonato, Stefania Corti, Megi Meneri

**Affiliations:** 1Dino Ferrari Centre, Neuroscience Section, Department of Pathophysiology and Transplantation (DEPT), University of Milan, Via Francesco Sforza 35, 20122 Milan, Italy; 2Stroke Unit, Neurology Unit, Neuroscience and Mental Health Department, Fondazione IRCCS Ca’ Granda Ospedale Maggiore Policlinico, 20122 Milan, Italy

**Keywords:** ischemic stroke, thrombi, clots, proteomics, metabolomics, transcriptomics, multiomic, large vessel occlusion, thrombectomy

## Abstract

Cerebrovascular diseases represent a leading cause of disability, morbidity, and death worldwide. In the last decade, the advances in endovascular procedures have not only improved acute ischemic stroke care but also conceded a thorough analysis of patients’ thrombi. Although early anatomopathological and immunohistochemical analyses have provided valuable insights into thrombus composition and its correlation with radiological features, response to reperfusion therapies, and stroke etiology, these results have been inconclusive so far. Recent studies applied single- or multi-omic approaches—such as proteomics, metabolomics, transcriptomics, or a combination of these—to investigate clot composition and stroke mechanisms, showing high predictive power. Particularly, one pilot studies showed that combined deep phenotyping of stroke thrombi may be superior to classic clinical predictors in defining stroke mechanisms. Small sample sizes, varying methodologies, and lack of adjustments for potential confounders still represent roadblocks to generalizing these findings. However, these techniques hold the potential to better investigate stroke-related thrombogenesis and select secondary prevention strategies, and to prompt the discovery of novel biomarkers and therapeutic targets. In this review, we summarize the most recent findings, overview current strengths and limitations, and present future perspectives in the field.

## 1. Introduction

The etiological diagnosis of acute ischemic stroke (AIS) subtypes is paramount to drive accurate secondary prevention strategies (such as anticoagulation in cardioembolic stroke—CE, associated with atrial fibrillation or antiplatelets in large artery atherosclerosis stroke—LAA), and to avoid recurrences. Undetermined stroke accounts for at least one-third of stroke patients [1], and up to 50% in certain subpopulations (e.g., cancer patients) [2], posing several challenges regarding secondary prevention.

In the last decade, the number of endovascular thrombectomy (EVT) interventions in patients with stroke and large vessel occlusions (LVOs) has dramatically increased following positive findings from crucial clinical trials [3], enabling histological, biochemical, and structural analysis of retrieved thrombi [4]. These analyses have correlated thrombi composition with histological and immunohistochemical methods with EVT recanalization rates, response to intravenous thrombolysis (IVT), radiological features, stroke severity, and functional outcomes. In addition, the cellular and molecular characteristics of cerebral thrombi are heterogeneous and provide information about their etiology [4]. Early studies investigated mainly red blood cells (RBCs), fibrin, and platelets [4,5], while more recent reports measured other components such as leukocytes, von Willebrand factor (VWF), and neutrophil extracellular traps (NETs) [6,7,8]. Particularly, two studies found that CE thrombi were richer in platelets than LAA thrombi [9,10], but others found the opposite [11,12]. In addition, although leukocytes may not be related to stroke etiology [13,14,15], some studies suggested a correlation between higher leukocyte content and a CE source [16,17]. Regarding VWF, this is present in all thrombi, ranging from 0.1 to 95% in concentration, independently of the origin [9,18,19]. Unfortunately, these findings have been largely inconclusive so far [20].

Recent studies [21,22,23,24,25,26] have evaluated omic techniques to assess thrombi composition. These refer to a variety of technologies enabling deep phenotyping of biological samples to define their molecular characteristics [27]. These approaches include proteomics, metabolomics, transcriptomics, or combinations of these. Though at early stages, these technologies may be instrumental to better defining stroke etiology [21], studying the mechanisms of thrombogenesis, and evaluating new molecular targets. In this topical review, we examine the most important associations between clot composition assessed using multi-omic technologies and stroke etiology and outcomes. We present the available studies, and we discuss current limitations and future perspectives. This work is based on all reports available (until December 2022) using omic analyses on stroke thrombi. For a more in-depth description of omic technologies in stroke and other analyses of thrombi, we refer to other recently published reviews [7,28].

## 2. Anatomopathological Studies

In the last few years, different studies have tried to correlate thrombus composition with stroke etiology and outcomes. Detailed histopathological and immunohistochemical analysis indicate that thrombus content in stroke is highly heterogeneous [12,18]. Clot composition typically consists of a variable amount of RBCs, platelet/fibirin (PF), leukocytes, bacteria, VWF, NETs, and extracellular DNA [5,7]. The architecture of RBC-rich clots has a limited complexity and is composed of RBCs densely packed within a thin fibrin network, and a small number of leukocytes. In contrast, PF-rich clots are more convoluted and contain VWF, WBCs, NETS, and extracellular DNA [7,29].

Several studies found an association between RBC-rich clots and LAA etiology while PF-rich clots have been related to cardioembolic mechanisms [13,14,16,30]. Recently, Brinjiki et al. obtained similar findings in a large multicenter study with thrombi from 1350 patients [30]. Conversely, other authors reported a correlation between RBC-rich clots and cardioembolic etiology [31]. Thus, these studies have been inconclusive so far to demonstrate the role of immunohistopathological analyses of clots in clinical practice.

Regarding undetermined etiologies, data from histopathological studies highlighted similar features between cryptogenic and cardioembolic clots, in particular, similar proportions of PF [18]. A recent meta-analysis of 21 studies found that cardioembolic and cryptogenic clots have a high PF content and indicated a positive association between RBC-rich clots and better reperfusion rates [32]. Accordingly, several studies showed that IVT and EVT are more effective in clots with higher content of RBCs [16,33,34]. Growing evidence pointed out that high content of NETs and WBCs of CE origin might be correlated with worse outcomes [20].

Although thrombus immunohistopathology has provided valuable data in stroke research, the wide variability of the findings currently hampers clinical translatability [20,32]. The implementation of more standardized methods and the complementary use of novel biomolecular techniques in future studies—such as omic technologies—may lead to more reliable results that could better guide therapeutic decisions.

## 3. Multiomic Studies

Data from multi-omic analysis on retrieved clots from patients with LVOs are scant and based on pilot studies with small sample sizes. Possible approaches include proteomic, metabolomic, and transcriptomic analyses. Currently, only two studies used a combination of methods (proteomic and metabolomic) [21]. All the other researches are based on single-method approaches (Table 1).

### 3.1. Proteomics

Proteomics refers to the large-scale analysis of the protein landscape in a biological entity at a specific time that is performed on bodily fluids and other biological materials in healthy individuals or patients with specific conditions, including stroke [28]. Since proteins are involved in most cellular processes, the composition of the proteome provides crucial information.

From a technical point of view, global and targeted mass spectrometry (MS) and antibody- and aptamer-based approaches have provided valuable results to study stroke proteomics [28]. These have been instrumental to investigate the composition of clots retrieved from patients with large vessel occlusions. Other authors have already published guidelines on blood and thrombi sampling for gene expression and proteomics analyses [39]. An exploratory proteomic study analyzed 20 thrombi from patients with AIS and correlated their content expressed in differentially expressed proteins (DEPs) with clinical features and laboratory parameters [25]. The concentration of 3 DEPs (septin-2, phosphoglycerate kinase-1, and integrin α-M) corresponded to the levels of serum low-density lipoprotein (LDL), while another peptide, septin-7, inversely correlated with the erythrocyte sedimentation rate. A smaller study assessed the proteome of 4 thrombi from AIS patients and found 341 out of 1600 proteins that were identified in all samples [24]. Bioinformatic analyses revealed clusters of proteins associated with immunological functions, cardiopathy-related proteins, and peripheral vascular processes. In addition, they identified four proteins (protein-glutamine gamma-glutamyltransferase 2, Actin α cardiac muscle 1, macrophage-capping protein, and putative elongation factor 1-α-like 3) that are specifically associated with platelet function and clots. Interaction network analyses suggested that some specific networks may be involved in clot formation, such as fibronectin 1 and 14-3-3 family of proteins and the TGFβ signaling. Fibronectin plays a role in platelet function [40] and can predict the hemorrhagic conversion of infarction and malignant ischemic stroke [41,42], whereas protein 14-3-3 is associated with platelet surface receptor and glycoprotein (GP) ib-IX-V complex-dependent signaling, which can initiate thrombus formation [43]. TGF-β is highly represented in platelets and is implicated in neuroinflammation after AIS [44]. Despite being tempered by the small sample size, the results from these studies demonstrate the feasibility of studying proteomics in clots from AIS patients.

Besides the descriptive characterization of the thrombus, proteomic analyses can inform on stroke etiology (Figure 1).

One important unanswered question is whether omic analyses would increase diagnostic accuracy compared with currently available radiological diagnostics and clinical scores. Darganzanli et al. compared 32 thrombi from patients with CE stroke vs. 28 with an LAA etiology using mass spectrometry and a machine learning method (support vector machine—SVM) [22]. Machine learning aimed to differentiate AIS etiologies based on specific subsets of proteins. They identified 438 out of 2455 proteins in all the analyzed samples, which clustered for key biological processes, such as metabolic pathways, cytokines assembly, leukocyte activation, migration, and cell adhesion. SVM highlighted that three proteins (eukaryotic translation initiation factor 2 subunit 3, Ras GTPase-activating-like protein IQGAP2, and coagulation factor XIII) classified correctly the 2 groups with an 88% accuracy. On univariate analysis, the coagulation factor XIII, the eukaryotic translation initiation factor 2 subunit 3, and the myosin light chain kinase levels were significantly different between the CE and LAA groups. When coupled with classic clinical predictors of CE etiology (e.g., history of cardiac failure), the predictive accuracy of the model raised to 97%. This study provided exploratory evidence suggesting that pathways associated with the clot-endothelium interaction may help to differentiate AIS etiologies.

Rossi et al. used mass spectrometry to compare 16 CE vs. 15 LAA thrombi in AIS patients [35]. They found 14 DEPs among 1581 identified proteins. LAA thrombi presented higher amounts of proteins involved in the ubiquitin-proteasome pathway, blood coagulation, or plasminogen-activating cascade. Clots from patients with CE strokes had more proteins involved in the ubiquitin-proteasome pathway, cytoskeletal remodeling of platelets, platelet adhesion by interaction with the VWF, and blood coagulation. However, adjusted analyses did not reveal significant differences between the two groups. The conservation of the samples in formalin instead of ice, a small sample size, no information on pre-stroke antithrombotic therapies, and the workflow time for stroke care could have hampered the generalizability of these data.

Abbasi et al. recently compared 25 CE vs. 23 LAA thrombi using proteomic techniques and applying pathway analyses and reverse-phase protein arrays to assess cellular interactions within the clots [36]. The analysis revealed strong interactions between PPAR-gamma, arginase-1, CD63, CD234, PKCαβ Thr 638/641, and VWF in CE clots, indicating that platelet signaling dominates in CE vs. LAA thrombi. The presence of multiple protein connections within inflammatory and immune cell proteins supports the concept of platelet-immune cell communication in CE clots. This concept is further reinforced by the enrichment of WBCs detected in the histological analysis of CE emboli.

The growing use of thrombectomy has also enabled the collection of intracranial blood samples during EVT. Comparing the protein content of intracranial and systemic blood can provide insights into the changes that occur in the brain during AIS. A pilot study using plasma proteome analysis found that several proteins, including ficolin-2, fetuin-B, prolyl endopeptidase fibroblast activation protein, uromodulin, and phospholipid transfer protein, were present at lower levels in intracranial blood compared to systemic arterial blood [45]. However, it is still unclear whether these differences are specific to stroke or simply reflect differences between blood sources. In a follow-up study, the authors found that elevated levels of VCAM1 in intracranial blood were positively associated with infarction and edema volume, indicating that VCAM1 may play a harmful role in stroke pathology [46]. VCAM1 is a molecule that links leukocytes to endothelial cells and has been linked to ischemic stroke pathologies [47]. Seminal studies on the topic have shown that cerebrovascular ischemia raises levels of leukocyte-endothelial adhesion molecules [48,49], which facilitates the adhesion and migration of inflammatory cells through blood vessels. Ischemic events worsen brain tissue damage by disrupting the endothelium and enabling leukocytes to migrate into the brain tissue [50]. Consistently with these results, protein clusters for key biological pathways in thrombi include leukocyte activation, migration, and cell adhesion [22]. These findings support the potential complementary value of coupling proteomic analyses of clots with that of intracranial blood sampled during EVT.

Overall, these proteomic studies are very preliminary for deriving generalizable conclusions. Inflammatory pathways and platelet function may be two key players in clot formation and changes in platelet-related pathways may provide clues to differentiating between CE and LAA etiologies.

### 3.2. Metabolomics

Metabolomics refers to the systematic investigation of the metabolites (i.e., small-molecule profiles) in biological samples, providing the molecular fingerprint of specific cellular processes [51,52].

Compared with proteomic techniques, metabolomics investigates smaller molecules (<2 kDa). The separation, precipitation, and removal of proteins are usually performed with simple extraction technologies, yielding metabolites. Recently, more focused analyses of metabolite subsets have resulted in new terms, such as “lipidomics”. This refers to the study of the variety of lipid species in eukaryotic cells that control cell membrane processes, and energy production and serves as precursors of bioactive molecules [53].

Available data on the use of metabolomic approaches on clots retrieved from AIS patients with LVOs are scant. So far, one study performed metabolomic analysis only, and 2 others metabolomic coupled with proteomics (combined proteomic and metabolomic paragraph). Early results on 5 thrombi indicate that the 10 most represented lipids were part of the phosphoethanolamine, phosphocholine, and fatty acids groups [23]. Plasma lipoproteins may play a role in thrombi formation through the interaction between ruptured atherosclerotic plaques and platelets [23]. In addition, patients with hypercholesterolemia present more frequently abnormal platelet function, such as alterations of lipoprotein-surface receptors interactions, increased platelet activation, and leukocyte recruitment [23,54]. Thus, the characterization of lipid composition in clots from AIS patients could help to better understand the role of different lipids in thrombogenesis, investigate the mechanisms of atherosclerotic plaque disruption in LAA-associated stroke, develop more tailored lipid-lowering and antithrombotic secondary prevention strategies for AIS.

#### Metabolomics and the Risk of AIS

Although metabolomics in clot research is still in its infancy, its broader use in AIS patients has offered new insights into predicting the risk of stroke. Sun et al. investigated the relationship between serum levels of 245 fasting metabolites and incident ischemic stroke in 3904 men and women [55]. Cox proportional hazard models were used to analyze the data and the results were validated in a separate sample of 114 stroke cases and 112 healthy controls. The results showed that levels of two long-chain dicarboxylic acids involved in the ω-oxidation of fatty acids, tetradecanedioate, and hexadecanedioate, were strongly correlated with CE strokes, independently of known risk factors. The findings suggest that these metabolites may be used as novel biomarkers for AIS and that pathways related to intracellular hexadecanedioate synthesis may play a role in stroke risk.

Similarly, two other studies found an association between altered levels of metabolites and the risk of AIS [56,57]. In patients at higher risk of stroke, Lee et al. reported lower levels of N6-acetyl-l-lysine, 5-aminopentanoate, cadaverine, 2-oxoglutarate, nicotinamide, l-valine, S-(2-methylpropionyl)-dihydrolipoamide-E and ubiquinone, and elevated levels of homocysteine sulfonic acid. Sensitivity analysis on patients with diabetes and smoking showed that these metabolites were specifically related to stroke, independently from potential confounders. Thus, lower lysine catabolites in patients at risk of stroke compared to controls support the idea of using these compounds as novel biomarkers for the early detection of stroke. Khan and colleagues compared 99 patients at risk of stroke and 301 non-risk controls and found that the former presented differential levels of 35 amino acids [57]. These included 10 metabolites, such as including L-tryptophan and homocysteine sulfinic acid, which were elevated in stroke patients at risk, providing evidence for the use of these compounds as biomarkers for early and non-invasive detection of AIS.

In a recent meta-analysis of 7 prospective cohorts, 10 metabolites seemed associated with a reduced risk of stroke. These included amino acids (such as histidine), high-density lipoprotein (HDL)2 cholesterol subfractions, pyruvate, and alpha-1-glycoprotein, a marker of acute phase response. Cholesterol in medium HDL and triglycerides in medium-large LDL were linked to overall stroke incidence, while phenylalanine and HDL subfractions (cholesterol and free cholesterol in HDL) were related to ischemic but not hemorrhagic stroke. Particularly, the strongest correlation was observed between histidine and stroke risk [58].

### 3.3. Combined Proteomic and Metabolomics

The combination of proteomic and metabolomic analysis on clots could help to better predict outcomes after AIS. In one study on 41 clots, 18 patients with favorable outcomes (modified Rankin Scale (mRS) score < 2) at 3 months had higher glucose and sorbitol levels; the latter was also an independent predictor of a good outcome. Though glucose is the substrate for sorbitol synthesis and is metabolized in the glycolytic pathway, this did not seem affected in these patients, as shown by proteomic analysis. Conversely, the excess of extracellular glucose in patients with AIS may be converted into sorbitol by the polyol pathway at stroke onset [26]. Regarding the predictive models, the authors applied the sparse Partial Least Squares-Discriminant Analysis (sPLS-DA) to determine the relationship between relevant metabolites and favorable clinical outcomes. After analyzing the loadings of features selected by the sPLS-DA model, they identified 20 variables per component for a total of 40 variables. The receiver operating characteristic (ROC) curve and Youden plot were then used to evaluate the association between major metabolites and the clinical outcome.

Recently, Suissa et al. used proteomic and metabolomic approaches in retrieved thrombi from AIS patients sampled consecutively to predict a CE vs. LAA etiology according to the ASCOD classification (A: atherosclerosis; S: small-vessel disease; C: cardiac pathology; O: other causes; D: dissection) [59]. Using liquid chromatography and mass spectrometry, the authors quantified a specific proteomic and metabolomic molecular signature in both groups and built a model with a significantly better predictive power (100% sensitivity/85.7% specificity) than classic clinical predictors of CE stroke (age, clinical severity at admission, and plasma levels of brain natriuretic peptide) [60,61]. In addition, they distinguished a peculiar proteomic signature in the 2 groups, with increased glycophorin-A (marker of red blood cells) and fibrinogen in CE strokes. Methodologically, the performance of classical predictors of CE and omic signatures was evaluated using ROC curves and area under curves receiver operating characteristic (AUROC). Similarly to the previously discussed study by Suissa et al. [26], the authors used sPLS-DA to discriminate between different origins of cerebral thrombi, applied logarithm transformations to the data, and analyzed untargeted proteomic and metabolomic results. The internal validity was checked using resampling methods and cross-validations. To externally validate each model, the authors assessed the prediction of new AF cases documented at 3-month follow-up in patients initially discharged as embolic strokes of undetermined source (ESUS). Finally, the predictive performance of both classical predictors and omic signatures was compared using AUROC pairwise comparison. Overall, this study and the one by Darganzanli (section on proteomics) provide evidence for the use of single or multiomic techniques to better predict stroke etiology in AIS patients, particularly regarding CE vs. LAA strokes.

Though valuable, this pilot study included a small and unbalanced number of thrombi per group (41 cardioembolic vs. 7 LAA), the predictive model did not consider radiological data and histopathological features, known as potential predictors of stroke etiology [4,62] and the authors did not check for consistency using other stroke subtype classification systems (e.g., TOAST, CCS). In addition, the model was based on 40 omic features, currently not feasible in clinical practice in many centers and the predictors of CE origin did not include electrocardiographic (ECG) data associated with atrial cardiomyopathy.

Atrial cardiomyopathy is a condition characterized by abnormalities in the structure and function of the atria and has been linked to a heightened risk of cardiovascular complications, independently from AF. The use of ECG parameters has become a crucial tool in predicting atrial cardiopathy. P wave parameters (PWPs) are an effective method for diagnosis, as they reveal underlying atrial structure, size, and electrical activation and can be quickly obtained through a standard 12-lead ECG [63,64]. PWPs include markers such as P wave duration, interatrial block, P wave terminal force in V1, P wave axis, P wave voltage, P wave area, and P wave dispersion, and can be combined to form a P wave index, such as the morphology-voltage-P-wave duration ECG risk score [65]. Abnormal PWPs may be linked with increased risks of AF, sudden cardiac death, and ischemic stroke in population-based cohort studies [63]. PWPs, either alone or in combination, may also improve the prediction of AF or ischemic stroke. Although further work is needed to standardize PWPs measurements, assess their accuracy and predictiveness, integrate novel techniques such as wavelet analysis and machine learning, and determine the benefits and risks of specific interventions for high-risk individuals, the widespread use of 12-lead ECGs coupled with thrombi analyses may help to improve the diagnosis, study, and treatment of CE stroke.

#### The Integration of Multiomic Techniques

The previously discussed findings suggest that the combination of multi-omic analyses may perform better than single omics and traditional clinical predictors in identifying stroke etiology. Particularly, the association of electrocardiographic and multi-omic analyses on thrombi may even provide further advancements in the field.

Overall, the combination of multi-omic technologies provides some advantages to understanding the complex and interconnected biological processes involved in AIS, including transcriptomic, proteomic, and metabolomic factors. These technologies will help in identifying new biomarkers, improving diagnosis accuracy and personalized medicine, and unraveling the underlying mechanisms of AIS. The integration of multiple data sets will lead to a better comprehension of the disease, enabling the discovery of new biological pathways and potential targets for therapeutic interventions. Additionally, multi-omics technology may improve the design and development of more effective drugs and interventions by providing a more detailed molecular characterization of AIS.

The so called “integromics”—i.e., the integration of multiple omic techniques—represents a further advancement in the field [66]. The process of integrating multiple omics starts by normalizing and reducing data to identify variations in patients with AIS; then the different types of molecular data are analyzed separately and finally combined to generate a comprehensive model [67]. The integration and analysis of multi-omics data aim to develop an efficient disease classification and prognosis model by exploring correlations and differences between the data, providing a foundation for precise and targeted patient treatment.

### 3.4. Transcriptomics

Transcriptomic refers to the use of next-generation sequencing (NGS) technology to obtain a complete set of RNA transcripts (RNA-seq) from different tissues [68]. This provides an accurate gene expression profile, offers more details on the cellular composition of samples, and enables the study of the molecular features involved in the various stages of the disease.

There are three main different techniques used for transcriptome analysis: bulk RNA seq, single-cell analysis (scRNA-seq), and spatial transcriptomics (sp RNA-seq). Although bulk RNA seq provides important data regarding the main molecular pathways underlying the disease, it fails to capture specific cell types and spatial information. To overcome the transcriptional diversity at the single-cell scale, scRNA-seq was developed for the first time in 2009 [68], enabling a deeper characterization of cellular heterogeneity in diseases, including cerebrovascular disorders [69]. SpRNA-seq is a novel technique that results in a gene expression matrix throughout tissue space. This methodology combines the strengths of bulk RNAseq and in situ hybridization, without requiring dissociation and manipulation of the sample, thus preserving the spatial organization of the tissues [70]. Currently, two commercially available platforms of spRNA-seq are available: 10X spatial transcriptomics from 10XGenomics and digital spatial profiler from NanoString Technologies.

The literature investigating the role of transcriptomic analysis in stroke research is scant. Current limitations include the quality of the retrieved material (e.g., number of cells), the collection, and the preservation of the samples. Most of the available studies were conducted on thrombi obtained during carotid endarterectomy and investigated not only the plaque but also the intima and the tunica media of the involved vessel [69]. So far, only Tutino et al. have performed bulk RNA seq in the clots of AIS. In 2021, they developed a new protocol to ensure sufficient RNA quality for further RNA seq analysis. In 73 clot samples retrieved by EVT, RNA seq analysis was feasible only in 48 thrombi, showing the limitations of this approach in terms of quantity and quality of samples and storage methods [37]. Recently, the same group studied the transcriptome profile of 38 thrombi identifying 174 differentially expressed genes (DEGs), 20 of which were shared by clots of different etiologies. RNA-seq highlighted with bioinformatic gene ontology analysis showed that CE clots express high levels of different genes involved in immunological processes (such as RANTES, PTGS1, MPO, MMP8, MMP9, C1QA, C1QB, NGAL, APOE, APOC1), suggesting an overall neutrophil, platelet, and innate immune system-dependent activation. Differently, LAA clots transcriptomes had greater enrichment of oxygen transport, oxidoreductase activity, and T-cell activation processes (with increased expression of HBD, HBA1/2, HBB, HBM, TRBC2, TRAC, IL7R, CCR7, CXCL5). Cryptogenic clots clustered in several profiles are potentially consistent with multiple concurrent etiologies [38].

Another study investigated specific pathways using quantitative real-time polymerase chain reaction (qRT-PCR) on clots [71]. Baek and colleagues explored the role of inflammation in stroke pathophysiology by measuring mRNA expression levels of interleukin (IL)-1β, IL-6, IL-8, IL-18, tumor necrosis factor (TNF)-α, matrix metallo-proteinase (MMP)-2, and MMP-9 in retrieved thrombi. In addition, they correlated these results with the presence of the susceptibility vessel sign (SVS), a radiological marker of paramagnetic content (e.g., deoxygenated hemoglobin), which is more frequently associated with an LAA etiology [72]. Out of 82 clots, 9 were associated with an LAA etiology and had a higher content of IL-1β. They demonstrated that patients lacking the SVS had higher expression of IL-1β, tumor necrosis factor-α, and matrix metalloproteinase-9 [71]. Altogether, these results suggest the value of combining analyses of transcription processes with radiological methods to better define stroke etiology.

The use of transcriptomic approaches could provide some advantages to studying clots in AIS patients. First, scRNA-seq could allow the study not only of cell-to-cell interactions but also of novel and rare cellular subtypes inside the clots. Second, spRNA-seq—by coupling extensive spatial barcoding of mRNAs on tissue with advanced imaging techniques and reconstructions—may provide unrivaled spatial resolution to distinguish different disease-relevant areas inside the thrombi, increasing the understanding of tissue heterogeneity and thrombogenesis. For instance, the two poles of a thrombus occluding an artery may show a varying composition according to the underlying etiology and different local microenvironments and potentially present a variable response to reperfusion therapies. Third, spRNA seq combined with other omic methodologies could achieve further steps forward in dissecting the composition of clots and thus increasing the knowledge of complex molecular mechanisms and allowing the discovery of new therapeutic targets in AIS.

## 4. Limitations and Perspectives of Omic Studies in Acute Ischemic Stroke

The use of omic approaches to study clots in AIS patients is still in its infancy and needs optimization. Current limitations include (1) the small size of study samples, (2) the lack of data on potential confounders (e.g., timing of administration, amount and type of thrombolytic therapy, pre-stroke antithrombotic therapies, workflow times, and stroke classification systems), (3) the heterogeneity of sampling procedures and study protocols, and (4) high costs. In addition, (5) the technology to perform such analyses is available only in selected comprehensive stroke centers, hampering the potential use in clinical practice, particularly in rural areas. The use of standardized protocols for sampling and analyzing data, the definition of shared guidelines, and the reduction in costs from improved technology could increase the use of omics in clinical practice (Figure 2).

Currently, omic analyses of clots are not routinely used in clinical practice. However, the possible future clinical impact of these analyses include: (I) identifying biomarkers for stroke mechanisms and prognosis, which could help clinicians to make more informed decisions about patient care; (II) better selecting antithrombotic treatments following the molecular characterization of clots, improving their efficacy; (III) evaluating the impact of risk factors such as genetics, lifestyle, and comorbidities on thrombosis and AIS; (IV) assessing the risk of recurrence based on the composition of clots, thereby opening up new avenues for personalized medicine for stroke patients in terms of better controlling vascular risk factors and selecting secondary prevention strategies.

Regarding future perspectives, a recent new tool in proteomic studies—Alphafold 2—represents a novel technological breakthrough in the field [73,74]. Alphafold 2 is a deep learning-based protein folding prediction algorithm developed by Deepmind. It uses artificial neural networks to predict the 3D structures of proteins from their amino acid composition with high accuracy, which is critical for understanding the functions of proteins and drug discovery. Alphafold 2 is trained on a large dataset of proteins and has achieved state-of-the-art results in predicting protein structures, outperforming traditional protein folding prediction methods. Although not tested on clots from AIS patients, Alphafold 2 may provide interesting results in stroke research. First, it could help in generating predictive models of protein structures in clots to provide novel insights into the molecular mechanisms of thrombosis. Second, it could improve the identification and quantification of proteins in clots, thereby increasing the understanding of thrombi composition and helping to select appropriate antithrombotic treatments. Third, it may support the integration of genomic, transcriptomic, and proteomic data to better understand the interplay between gene expression and protein synthesis leading to thrombi formation. Fourth, Alphafold may offer novel insights into the post-translational modifications of proteins in clots, offering new perspectives into the regulation of thrombosis. Fifth, it may become instrumental in modeling the effects of drugs acting on protein-protein interactions in clots to expand treatment options and identify new therapeutic targets.

Incorporating machine learning algorithms into the analysis of omic data will provide further advancements in the field [66]. Particularly, machine learning can enhance the analysis of omic data by allowing the identification of complex patterns and relationships within the data, beyond what is possible through traditional statistical methods. This can improve the accuracy of disease classification, prognosis, recurrence risk modeling, and drug response prediction. Machine learning algorithms, such as decision trees, random forests, and artificial neural networks, can be trained on large omic datasets to identify significant features and predict outcomes with high accuracy [75]. In addition, machine learning can enable the integration of multiple omics data types, such as genomics, transcriptomics, proteomics, and metabolomics, to generate more comprehensive models of disease mechanisms. This can lead to a deeper understanding of the underlying molecular mechanisms of AIS and inform the development of personalized therapeutic strategies.

Despite being still limited to comprehensive academic centers, we believe that in the era of precision medicine and individualized treatments, it is mandatory to encourage further research in this field with prospective studies.

## Figures and Tables

**Figure 1 ijms-24-03419-f001:**
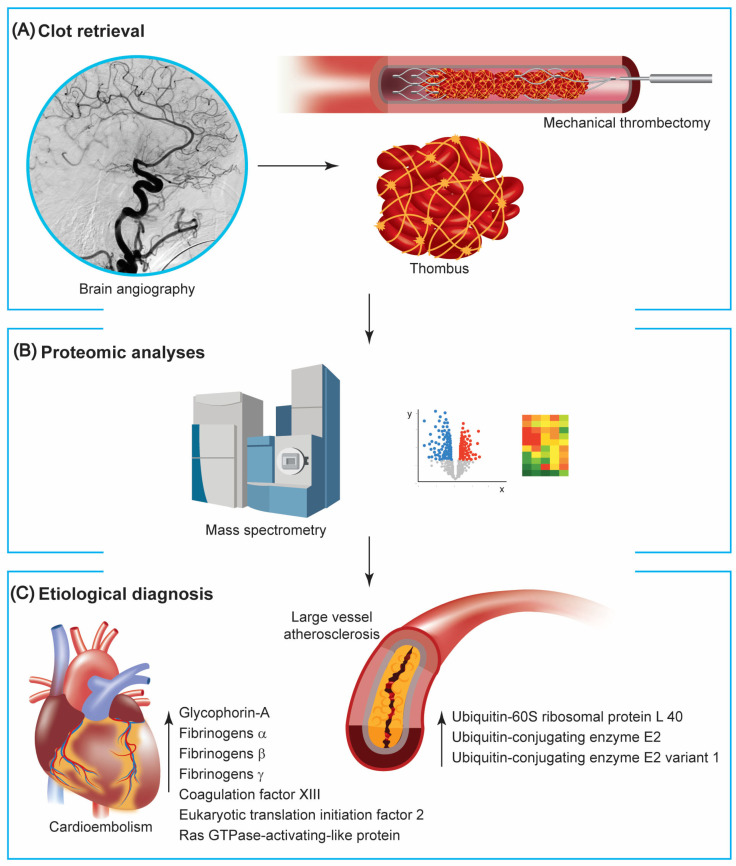
Proteomic studies on clots retrieved from mechanical thrombectomy. (**A**) Patients with large vessel occlusion can be treated with mechanical thrombectomy, making the retrieved thrombus available for further analysis. (**B**) Mass spectrometry and pathway analyses of generated data are instrumental to investigate proteins extracted from clots. (**C**) Available reports using proteomic technologies point out a different composition between clots with cardioembolic and atherosclerotic origins, with the formers containing more proteins associated with red blood cells and the coagulation cascade and the latter with the proteasome-ubiquitin pathway.

**Figure 2 ijms-24-03419-f002:**
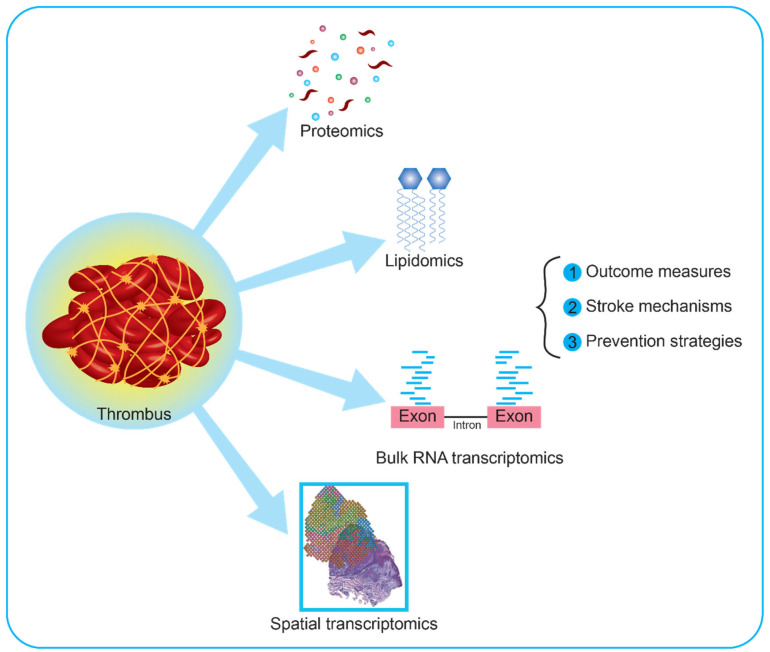
Current application and future perspectives on clot research using multi-omic approaches. Thrombi retrieved from patients with large vessel occlusions can be analyzed using proteomic, metabolomic, and transcriptomic approaches, or combinations of these. Recently, metabolomic studies have focused on specific subtypes of metabolites, i.e., lipids. These are involved in cell membrane dynamics, transport of bioactive macromolecules, and molecular synthesis, holding the potential to increase knowledge on the cellular composition of clots. Regarding transcriptomic analyses, available reports both on human and animal models used bulk RNA, namely the pooled RNA derived from all the cells in a sample. Further interesting approaches would include single-cell RNA and spatial transcriptomics. The former investigates the whole transcripts of each cell in the clot, while the latter provides a spatial map based on barcoded RNAs with high spatial resolution, enabling the investigation of rare cell types and new molecular targets.

**Table 1 ijms-24-03419-t001:** Published omic studies on clots retrieved from patients with large vessel occlusion.

Technique	Author	Sample Size	Goal of the Study	Main Findings
Proteomics	Rao et al. [25]	20	To identify and correlate clot DEPs ^1^ with clinical features.	Septin-2, phosphoglycerate kinase-1, integrin α-M present in clots from patients with high LDL.
	Munoz et al. [24]	4	DEPs characterization of the clot.	342 DEPs clustered with immunological, cardiovascular, and platelet function processes.
	Darganzanli et al. [22]	60	To identify and correlate DEPs clot with AIS etiology.	438 DEPs clustered according to metabolic pathways, cell adhesion, leukocyte activation, and migration. Clot-endothelium interaction pathway predicts etiology. Coagulation Factor XIII levels are higher in CE clots.
	Rossi et al. [35]	31	To identify and correlate DEPs clot with AIS etiology.	14 out of 1581 DEPs involved in distinct pathways differ between LAA and CE etiologies (statistically non-significant.)
	Abbasi et al. [36]	48	Protein signatures correlate with AIS etiology.	Platelet signaling, in particular platelet-immune cell communication, prevails in CE clots.
Metabolomics	Martha et al. [23]	5	To identify lipid clot profile.	Glycerophospholipid and fatty acids as the most represented lipids.
Transcriptomics	Tutino et al. [37]	73	To establish a protocol for RNA seq on clots.	Only 48 out of 73 clots were available for informative RNA sequencing.
	Tutino et al. [38]	38	To assess clot gene expression and AIS etiology.	CE clots presented higher expression of genesinvolved in neutrophil activity, platelet function, and innate immune system activation processes. LA clots presented higher expression of genes involved in T cell-mediated processes and oxidoreductase activity.
Combined (Proteomics + Metabolomics)	Suissa et al. [21]	48	To predict AIS etiology by multi-omic analyses.	Combined proteomic and metabolomic clot profiles have significant predictive power (100%sensitivity) of CE etiology.
	Suissa et al. [26]	41	Multi-omic profile of clots and correlation with outcomes.	Multi-omic profile of clots and association with outcomes.

^1^ DEPs (differentially expressed proteins).

## Data Availability

No new data were created or analyzed in this study. Data sharing is not applicable to this article.

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
