# Peer review of "Advancing Stroke Research on Cerebral Thrombi with Omic Technologies"

_ijms, 2023, doi:10.3390/ijms24043419_

Round 1

Reviewer 1 Report

A competent and review into a new and burgeoning field of omics in ischemic stroke. This paper is well written and I have only a few comments:

The classic clinical predictors of cardioembolic stroke specified are out of date. age, NIHSS , and BNP are useful. However there are many ECG parameters that look at atrial cardiopathy which are currently in use.

The authors can mention the published existing protocols established to give a framework for uniform sample collection: Fraser JF, Collier LA, Gorman AA, Martha SR, Salmeron KE, Trout AL, Edwards DN, Davis SM, Lukins DE, Alhajeri A, Grupke S, Roberts JM, Bix GJ, Pennypacker KR. The Blood And Clot Thrombectomy Registry And Collaboration (BACTRAC) protocol: novel method for evaluating human stroke. J Neurointerv Surg. 2019 Mar;11(3):265-270. doi: 10.1136/neurintsurg-2018-014118.

There are some papers which the authors have missed:

Abbasi M, Fitzgerald S, Ayers-Ringler J, et al. (February 22, 2021) Proteomic Analysis of Cardioembolic and Large Artery Atherosclerotic Clots Using Reverse Phase Protein Array Technology Reveals Key Cellular Interactions Within Clot Microenvironments. Cureus 13(2): e13499. DOI 10.7759/cureus.13499

The authors could also compare the wider field of omics In blood samples (especially intracranial blood samples) with the output that is currently available in these few clot studies to look for similarities.

Reviewer 2 Report

The authors reviewed recent studies applied single- or multi-omic approaches - such as proteomics, metabolomics, transcriptomics, or a combination of these to investigate clot composition and stroke mechanisms, showing high predictive power. Particularly, one pilot studies showed that combined deep phenotyping of stroke thrombi may be superior to classic clinical predictors in defining stroke mechanisms. The manuscript is well structured, and the procedures are logically sound. However, I do have some questions with regard to the manuscript:

1. In section 3, the authors reviewed main findings of several multiomic studied. Some studies used multiomic profile to predict stroke etiology and outcomes. The prediction methods should also be reviewed and compared. 

2. Is there any clinical application or impact of these studies?

3. Is stroke mechanism or treatment more clear according to these studies?

4. In proteomics studies, new tools, such as Alphafold 2, have been developed. Is there any new application in stroke researches?

5. Section 2.1-2.4 should be 3.1-3.4.

Reviewer 3 Report

In this review, Costamagna et al. summarized the most recent findings of multi-omic studies on stroke-related thrombogenesis. They also reviewed current strengths and limitations and presented future perspectives. It is a timely and significant review in the field.

I have the following suggestions to improve the manuscript.

The authors should discuss the specific application of metabolomic technology in AIS. For example, can it predict the risk of AIS?

In Section 2.2, the authors introduced relevant studies on metabolomics, but the authors need a more profound discussion in this section.   

At the end of the multi-omics section, the authors can better summarize the advantages and significance of multi-omics technology for stroke research, development, and clinical diagnosis.

The authors need to discuss omic technologies' future development and research direction in the limitations and perspective section. For example, can other advanced techniques be combined to promote clinical translatability?  

There are problems with punctuation in the text and the table. For example, the authors should add a period at the end of the first paragraph of Section 2.4. Finally, the authors should correct grammar and syntax errors throughout the main text and table 1, if any. 

Round 2

Reviewer 2 Report

All questions were answered properly.

Reviewer 3 Report

my concerns have been addressed